



# Ongoing grounding line retreat and fracturation initiated at the Petermann Glacier ice shelf, Greenland after 2016

Romain Millan[1,2], Jeremie Mouginot[2,3], Anna Derkacheva[1], Eric Rignot[3,4], Pietro Milillo[5], Enrico Ciraci[4],

Luigi Dini[6], Anders Bjørk[2]

[1]Université Grenoble Alpes, CNRS, IRD, INP, 38400, Grenoble, Isère, France
[2]Department of Geosciences and Natural Resources Management, University of Copenhagen, 1350, Copenhagen, Denmark
[3]Department of Earth System Science, University of California, Irvine, 92697, CA, USA
[4]Jet Propulsion Laboratory, Caltech, Pasadena, CA, USA
[5]Department of Civil and Environmental Engineering, University of Houston, TX, USA
[6]Italian Space Agency, Matera, Italy

*Correspondence to*: Romain Millan (romain.millan@univ-grenoble-alpes.fr)

**Abstract.** The Petermann ice shelf is one of the largest in Greenland, buttressing 4% of the total ice sheet discharge, and is considered dynamically stable. In this study, we use differential synthetic aperture radar interferometry to reconstruct the grounding line migration between 1992 and 2021. Over the last thirty years, we find that the grounding line of Petermann retreated 4 km, 7.5 km and 4.5 km in the western, central and eastern sectors, respectively. However, it is only since 2017 that the glacier has undergone a significant retreat in its central section, 20   receding more than 5 km along a retrograde bed grounded 500 m below sea level. Simultaneously, two large fractures developed, splitting the ice shelf in three sections, with partially decoupled flow regime. The retreat followed the warming of the ocean waters by 0.4ºC in Nares Strait. As a result, the glacier sped up by 15% in 2016-2018. While the central sector stabilized on a sill, the eastern flank is sitting on top of a down-slopping bed, which might accentuate the glacier retreat in the coming years.

## 1 Introduction

The Greenland Ice Sheet (GrIS) is a main contributor to sea level rise at present (Reager et al. 2016, Mouginot et al., 2019, Shepherd et al. 2020), and its contribution is expected to increase in the coming years (e.g., Choi et al., 2021). While the vast majority of Greenland marine terminating glaciers do not end in extensive floating extensions, ~18% of Greenland's total ice volume is buttressed by ice shelves in the North (Mouginot et al., 2019, Morlighem et al., 2017). It is of prime importance to 30   document the evolution of these ice shelves as their weakening may destabilize this sector and significantly increase the contribution of GrIS to sea level rise (Mouginot et al., 2014).



Petermann glacier is located on the northwestern part of the GrIS and ends in the second largest floating extension after Nioghalfjerdfjorden glacier, with a length of 70 km in the 1990's and a width of 20 km. In the last 20 years, this glacier has increased its ice discharge from 10.5 Gt/yr up to 11.5 Gt/yr (Mouginot et al., 2019). Recent studies using Worldview Digital
Elevation Models (DEMs) revealed submarine melt rates near the glacier grounding line that exceed 50 m/yr (Wilson et al., 2017). The grounding line is the most sensitive region of a glacier as its migration significantly modulates resistance to flow (Fürst et al., 2016; Reese et al., 2018; Thomas 1979). Satellite radar interferometry data from the Earth Remote Sensing Satellite-1 were used to map the grounding line of Petermann Glacier in the 1990s and detect its migration between 1992 and 2011 (Rignot et al., 1996; Hogg et al., 2016). From these observations, it was concluded that the long-term changes in
grounding line locations on Petermann Glacier were not detectable because of tidal modulations of the position and despite the reported ice shelf thinning (Wilson et al., 2017; Rückamp et al., 2019; Münchow et al., 2014). Petermann Glacier also lost about half of its floating extension via two large calving events in 2010 and 2012. The glacier showed a limited velocity response to these calving events (Hill et al., 2018; Rückamp et al., 2019), suggesting that the glacier was still dynamically stable. Hill et al., (2018) suggested however that if future calving occurs within 12 km of the grounding line, the changes in
the ice-shelf buttressing would be sufficient for the glacier to accelerate substantially.

Since 2014, the launches of a constellation of two Sentinel-1 (S1) C-band synthetic aperture radar satellites by ESA for the European Copernicus initiative have increased the perspectives to study glacier dynamics and grounding line migration (Milillo et al., 2019; Mohajerani et al., 2021). Following recommendations by the Polar Space Task Group under the umbrella of the
World Meteorological Organization, Sentinel-1 has continuously acquired since June 2015 a set of six tracks that cover the entire coast of Greenland, hence providing a systematic and comprehensive coverage at a revisit time of 6 days. This continuous observation record has made it possible to map the grounding lines of ice shelves in a more comprehensive and systematic way. Indeed, Sentinel-1 allows not only to map the grounding line again since ERS-1, but on multiple occasions, several times a year, hence providing information about the zone of short-term migration or grounding zone (Mohajerani et al., 2021).
Following a first set of data in 2013, the Agenzia Spaziale Italiana's Cosmo-SkyMed® constellation has acquired 1-day repeat pass data over Petermann Glacier in 2020, providing additional coverage of its grounding line migration in addition to Sentinel-1a/b.

Here, we present the grounding line history of Petermann glacier, spanning between 1992 and 2021, or 30 years. We combine
satellite radar interferometry and optical imagery to track both the grounding line and the change in ice velocity of the glacier. We examine concurrent changes in the ocean waters surrounding the glacier. We conclude on the evolution of the glacier grounding line and its consequences for the future years.



## 2 Data and methods

### 2.1 Interferometry

We use Interferometric Synthetic Aperture Radar (InSAR) data from the ESA's Earth Remote Sensing radar satellite (ERS-1) acquired in 1992 and 2011 with a 3-day revisit time, in 1995/1996 from the ERS-1 and ERS-2 tandem mission, and in 2011 from a 3-day revisit time ERS-2 mission end. Data are downloaded via the ESA Online Dissemination Service as single look complex scenes and processed using the GAMMA Software (Werner et al., 2000). We measure the tide-induced vertical motion of ice using a Quadruple Differential SAR Interferometry approach (QDInSAR) (Rignot et al., 2011). In order to

maintain good phase coherence in fast flowing regions, we first coregister single look complex (SLC) data by estimating the pixel offset using classical speckle tracking technique (Michel and Rignot., 1999; Scheuchl et al., 2016). Interferograms are then assembled by calculating the phase difference between the two co-registered SLCs. To obtain the grounding line position, we combine two interferograms spanning the same time interval, correct for topography, and differentiate them (Rignot et al., 2011). We use the GIMP v1 DEM time-tagged in 2007 to remove the topographic signal. For the time period 2014 to 2021,

we use observations collected by Sentinel-1 with a repeat cycle of 6 to 12 days in Interferometric Wide Swath (IW) mode, a Terrain Observation with Progressive Scans SAR (TOPS) mode. In order to avoid phase jumps at burst boundaries, interferograms from this sensor are processed using a precise TOPS coregistration methods, which also registers the Doppler history of the slave data (Scheuchl et al., 2016). We use the GIMP DEM v2 time-tagged in 2014 to correct for the topographic phase for S1 (Howat et al., 2017). For both ERS-1 and S1, the differential interferograms are formed after geocoding the

interferograms in geographic coordinates (North Polar Stereographic projection) at 25 meters posting. As in Rignot et al. (2014), we map the inward limit of detection of vertical motion, where the glacier first lifts off its bed.

COSMO-SkyMed® (CSK) grounding line measurements use two satellites (CSK2 and CKS4) each with a 16-day repeat cycle. The temporal baseline between CSK2 and CSK4 is one day. In order to avoid changes in the horizontal velocity of the glacier

we combine a double-difference interferogram using two one-day interferograms acquired 16 days apart. CSK acquires in STRIPMAP mode three consecutives frames to cover the entire glacier and its ice shelf. We stitch all the frames in order to combine a single single-look-complex (SLC) covering a 40x120 km swath. Following the approach described in Milillo et al., (2017) and Brancato et al., (2020), we apply 8 looks in both range and azimuth to improve InSAR phase coherence. The final geocoded product has a resolution of $25 \times 25$ m. As for the Sentinel-1 case, we use the GIMP v1 DEM time-tagged in 2007 to

remove the topographic signal. Finally, the grounding line is characterized using QDInSAR with the interferograms assembled over the four epochs, in the vicinity of the flexure zone, as for ERS-1/2 and Sentinel-1 data (see above). CSK completes the S1 observations in 2020 and 2021, by providing some high-resolution observations of the grounding line.





## 2.2 Ice velocity

In addition to the migration of the grounding line, we monitor the evolution of glacier velocity by calculating the surface
displacement from 3 different satellite sensors from images collected between 2013 and 2021. Two of them, ESA's Sentinel-
2 (S2) and NASA's Landsat-8 (L8), are optical imagers and one, ESA's Sentinel-1 (S1), is a synthetic aperture radar operating
in C-band. We use persistent surface features or speckle to map ice displacements between two consecutive images. We
calculate the normalized cross-correlations between the reference and slave image chips using repeat cycles shorter than 30
days for Landsat-7/8 and Sentinel-2, and 12 days for Sentinel-1 (Mouginot et al., 2017; Millan et al., 2019; 2022). Between
1999 and 2012 we supplemented our Landsat-7 ice velocity record with repeat cycles ranging from 336 to 400 days. For L8,
S2, and S1, sub-images of $32 \times 32$, $32 \times 32$, and $192 \times 48$ pixels are used respectively. We calibrated our displacement maps
by taking advantage of the ice velocity products from prior surveys in Mouginot et al. (2017). The final calibrated maps are
resampled to 150 m posting in the north polar stereographic projection (EPSG:3413). The time series established is completed
by historical measurements made from ERS-½, RADARSAT-1, ALOS/PALSAR, ENVISAT/ASAR, Landsat 4 to 7 and
TerraSAR-X (Mouginot et al. 2019, Joughin and Howat). The satellite-derived measurements between 2015 and 2021 are
post-processed with locally weighted polynomial regression (Derkacheva et al., 2019) to improve signal to noise ratio and data
redundancy. Relative changes in velocity are measured from the 2014-2015 winter mean velocity (January to March).

In order to monitor the propagations of rifts and fractures, we derive the evolution of the shear strain rate for 2000 and 2019
using annual ice velocity mosaics (Mouginot et al. 2019). Strain rates were retrieved using the same methodology as described
in Alley et al., 2018, where the shear strain rate is defined as :

$$\dot{\varepsilon}_{shear} = (\dot{\varepsilon}_y - \dot{\varepsilon}_x) \cos\alpha \sin\alpha + \dot{\varepsilon}_{xy}(\cos^2\alpha - \sin^2\alpha)$$

where $\dot{\varepsilon}_x$, $\dot{\varepsilon}_y$ and $\dot{\varepsilon}_{xy}$ are the component of the strain rate tensor, calculated from the two components of the ice velocity field
(Nye, 1959), and $\alpha$ is the flow angle, defined counter-clockwise from the x-axis (positive in the x direction) (Alley et al.,
2016).

## 3 Results

### 3.1 Grounding line evolution

Overall, we formed more than 800 quadruple difference interferograms, with 90% from S1 while the rest was acquired from
ERS and CSK, which allows grounding line monitoring at a high temporal frequency. In the summer, decorrelation due to
surface melting prevents us from mapping the grounding line on most interferograms. Differential interferograms and
grounding lines are shown in Figure 2 for 1992, 1996, 2011, 2015, 2016 and every year from 2018 to 2021. For every single



year, we display all the grounding line locations, which allows us to document its interannual spatial variability caused, for
example, by oceanic tides.

In 1992, the grounding line position of Petermann glacier was located 70 km from the ice front and remained relatively stable
until 2011 within uncertainties, which are about 1 km (Fig 2, 3). Between 2011 and 2016, the western (T1) and eastern (T4)
margins of the grounding line retreated by 3 km. During the same time period, we observe a retreat of 1.1 km and 2 km along
transects T2 and T3, respectively (Fig 2 & 3). In 2015-2017, we were only able to map the central and eastern portion of the
grounding line. The high number of ERS and Sentinel-1 double difference interferograms allows us to provide a rough
quantification of the grounding zone width of the glacier, which averaged 550 m in 1992 and 630 m in 2015, before the retreat
(Fig 3a). These estimates are consistent with those of Hogg et al. (2016) with 470 m in 1992-2011.

Between 2017 and 2019, the grounding line position along the central transect T2 retreated by more than 4 km, while the rest
of the grounding line remained relatively stable. Between 2019 and 2021, the retreat continued along T2 and the glacier
retreated by another km (Fig 2 & 3), for a total retreat of 7 km compared to 1992. Within the same timeframe, the eastern (T4)
and western (T1) sections of the grounding line retreated by 2 km, for a total retreat of 3.8 km and 4 km respectively, since
1992.

**3.2 Ice shelf fracturation**

In addition to the grounding line signature in the differential interferograms, we note the formation between 1992 and 1996 of
a decorrelation structure in the eastern side of the shelf parallel to the flow direction (Fig 2a), which seems associated with the
development of fracture zones. The fracture zones are manifest in the interferograms as regions of abrupt discontinuity in
phase, indicating that the two sides of the fractures are not flexing exactly at the same pace, which suggests a fracture across
the entire thickness of the floating tongue, rather than just a surface crack. In 2011, another parallel fracture zone is visible in
the western end of the ice shelf (Fig 2d-e). Between 2016 and present, these features became more pronounced and now extend
on both sides of the portion of the grounding line that retreated (Fig 2e-i). The surface manifestation of these fracture is also
visible using optical imagery from Landsat 7-8 (Fig 4c-d), with two distinct, and large crevasses developing on the shelf. Using
insights from calculated strain rates, it is clearly visible that the development of these fractures corresponds to regions of
particularly high shear (Fig 4a,b). Indeed, while the shear along the eastern fracture averaged -0.02 yr$^{-1}$ in 2000, it doubled to
-0.04 yr$^{-1}$ in 2019. On the other hand, the fracture on the western side of the grounding line is a region of particularly high
shear with a measured shear strain rate of up to -0.08 yr$^{-1}$ in 2019 (Fig 4b).

**3.3 Time series of ice velocity**

In Fig. 3b, we show the evolution of Petermann Glacier's velocity at a location 1 km upstream of its 2021 grounding line for
the time period 1984 to 2021. After 2014, we display in Fig. 3c the surface velocity across a 75 km long profile (A-A') that



coincides with transect T2 (cf Fig 1). We find that, between 1984 and 2021, the average speed has increased by 14%, or 150 m/yr (Fig. 3a). The glacier motion was fairly stable until 2014. A pronounced speedup started after the summer of 2015, with a speed of 1,100 m/yr, and ended in 2018 at a speed of 1,200 m/yr. Superimposed on this interannual trend, the frequent velocity measurements available after 2013 reveal a strong summer acceleration of 10 to 15 % compared to the winter speed

(Fig. 3b). The summer acceleration occurs uniformly along the ice shelf and propagates 40 km upstream of the grounding line (Fig. 3e). The signal may even extend further inland, but it is not distinguishable from noise in our observations. No delay between the acceleration at the grounding line and 40 km upstream is detectable on the weekly time series. The amplitude and duration of these summer speed-ups varies between years and is correlated with the intensity and duration of the melt season.

We assembled a time series of surface flow velocity across three gates along the glacier width (Fig 1). The profile at the grounding displays a pronounced asymmetry in glacier flow, with ice velocity of 1,000 m/yr in the west (km 0 to 10 of Figure 3c) vs ice speed >1,300 m/yr in the eastern side of the grounding line (>km 10 of Figure 3c). Along the ice shelf, the lateral difference in speed diminishes rapidly and gives way to an ice shelf with a surface velocity ranging between 1,000 m/yr close to the margins and >1,300 m/yr in the central part. Interestingly, the velocity profiles consistently show two steep transitions

at km 5 and km 12 of Fig 3d, where the ice velocity abruptly increases from 1,100 m/yr to more than 1,300 m/yr over a distance of ~3 km. These transitions are detected more than 30 km downstream of the grounding line, and are more pronounced after 2016 (Fig 4d).

### 3.4 Ocean thermal forcing

We reconstruct the history of ocean thermal forcing by compiling Conductivity Temperature Depth (CTD) measurements from

the Hadley center (bodc.ac.uk) within Petermann fjord and Nares Straits, spanning from 1960 to 2019, combined with CTD from NASA's Ocean Melting Greenland Earth Venture Sub-orbital mission from 2016-2021 (Fenty et al., 2016). A warming signal was detected at depth from the 1970s to the 2000s and an even stronger signal has taken place in the last decade (Fig 5). Overall ocean temperature in the fjord at 350-450 m depth (the maximum grounding line depth is 500 m) is 0.4°C warmer in the 2000s than in the 1970s-1980s. This warming signal has been documented elsewhere (Münchow et al., 2018) with a change

in temperature of 0.23°C between 2003 and 2009 (Washam et al., 2018).

### 4 Discussion

Our results indicate that the surface velocity has remained relatively constant between 1986 and 2010 (Fig 4a). The significant glacier speed-up that began in 2016 seems to precede the major episode of grounding line retreat in 2017 observed along transect T2 (Fig 3). After 2018, the surface velocity of Petermann remained stable, at 1200 m/yr (Fig 3b), while the grounding

line continued to retreat rapidly (Fig 3a). This timing coincides with the classical sequences of events leading to the destabilization of ice shelves, with a reduction in buttressing due to enhanced basal melting by a warmer ocean triggers an





increase in ice flow at the grounding line, which in turn leads to enhanced ice shelf thinning that causes the grounding line to further retreat (Schoof., 2007; Rignot et al., 2014; Pattyn and Morlighem., 2020). Furthermore, the pronounced abrupt transitions in ice flow velocity (Fig. 3c-d) are consistent with the development of breakup zones along the ice shelf length (Fig 4), showing a partially-decoupled flow regime between different parts of the ice shelf. The rapid and more pronounced development of these fracture zones is consistent with the timing of the grounding line retreat after 2016. While the ice flow of Petermann has not yet increased as much as other destabilised glaciers in Greenland such as Jakobshavn Isbræ (Motyka et al., 2011) or Zachariae Isstrøm (Mouginot et al., 2015), these results may suggest that the Petermann may be entering a phase of destabilization and weakening of its ice shelf.

The warming ocean signal precedes the increase in speed and grounding line retreat (Fig. 3-5). A warmer ocean will erode the ice shelf faster because of the enhanced undercutting at the glacier grounding line (An et al., 2021). A calculation of the rate of undercutting based on the model of Rignot et al., (2016) for a grounding line depth of 500 m, neglecting the role of subglacial water fluxes, suggest that the rate of undercutting, $q_m$, should have increased by 15% as a result of the warming, which may not be sufficient to explain the retreat. Indeed, an increase in yearly undercutting from 140 m/yr to 180 m/yr only forces a retreat of the grounding line of 40 m/yr. Glacier thinning however also contributes to grounding line retreat since a thinner glacier reaches flotation sooner. With dynamic thinning at about 1 m/yr (Smith et al,. 2020) on a -0.11% bed slope and a surface slope of 0.8% measured along the profile C-C', we calculate an expected retreat of about 130 m/yr (Rignot et al. 2016; Hogg et al., 2016). Combining the two effects, we are still below the retreat of 1.4 km/yr observed locally between 2017 and 2021 (Fig. 3a), hence the retreat at the glacier center may not be only conditioned by glacier dynamic thinning and enhanced undercutting by warmer ocean temperature. Indeed, other processes may have influenced the retreat of Petermann Glacier, such as the increase of subglacial water discharge (runoff) that would enhance basal melt at the grounding line (Jenkins et al., 2011), or the intrusion of pressurized seawater in newly formed cavities as proposed by Milillo et al., 2019.

A comparison of our 30-year long time series of grounding line retreat of Petermann Glacier with BedMachine v3 shows that the glacier retreat is correlated with the bedrock geometry. Indeed, the recent grounding line migration (>7 km between 1992-2021) coincides with a retrograde bed slope that deepens from 470 m to more than 517 m below sea level (Fig 1b, green rectangle). This pattern of retreat is similar to what has been observed for the Thwaites glacier in the Amundsen Sea embayment, where the retreat proceeded faster along topographic depressions of the bedrock (Milillo et al., 2019). In contrast, the eastern and western portions of the grounding zone have migrated slower than the central part, probably because of slightly prograde bedrock at these locations (Fig 1b). The eastern portion of the grounding line has remained stable since 2016, on a high rise in the bedrock grounded 490 m below sea level. However, we note that the bedrock deepens over the next 8 km on a retrograde slope down to 540 m depth (Fig 1c), which may accentuate the glacier retreat in the coming years (Schoof et al., 2007).

## 5 Conclusion

In this study, we use SAR interferometry to reconstruct the history of Petermann's grounding line over the last 30 years. For the first time and quite recently, a significant and rapid retreat of its grounding line is observed. The central section of the grounding line is now more than 7 km upstream of the 1992 position. This recession initiated after 2017 along a retrograde bed and followed a rapid rise in ocean temperature in the Nares Strait. Within the same timeframe, large fractures split-up the ice shelf in three sections, with high strain rates, and a partially decoupled flowing regime. In 2016, the glacier surface flow velocity increased markedly for the first time in the last few decades before stabilizing in 2018, while the grounding line continued to retreat up to present. Large sections of the grounding line sitting on down-sloping beds could retreat further in the coming years.

## Data availability

CTD data are available at https://climatedataguide.ucar.edu/ . Data will be permanently archived through a public data portal upon acceptance of the paper. Synthetic Aperture Radar data are freely available at https://earth.esa.int/eogateway.

## Author contribution

R.M and J.M conceived and designed the research. R.M, J.M, E.C processed and analysed data. A.D processed ice velocity time series. E.R processed and analysed ocean data. All authors participated in the writing of the manuscript.

## Acknowledgements

We acknowledge support from the French National Research Agency (ANR) grant (ANR-19-CE01-0011-01). We thank the European Space Agency for the use of ERS and Sentinel-1 acquisitions, and the Italian Space Agency (ASI) for granting the use of CSK products in the framework of the "Ghiacciai" project.

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




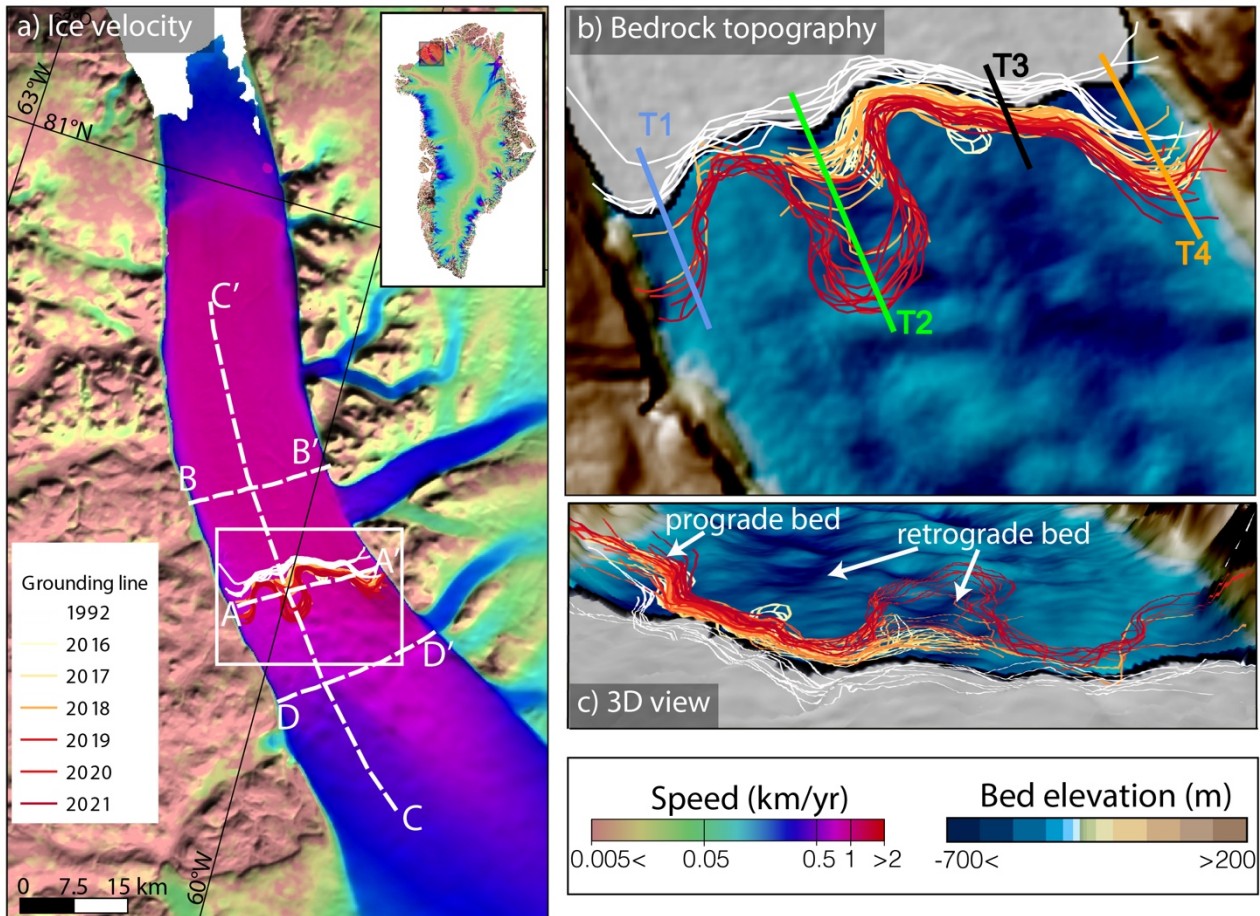

**Figure 1.** (a) Satellite derived surface flow velocity of Petermann Glacier's ice shelf colour coded on a logarithmic scale from brown to red. Dashed white line indicates the location of velocity profiles (c.f. Fig 3). Inset map shows the location of the Petermann over the ice velocity map of the Greenland ice sheet (Mouginot et al., 2019). (b) Bedrock geometry from BedMachine v3 on a blue colour scale (Morlighem et al., 2017). Lines T1, T2, T3 and T4 are used in Fig. 3a to show the evolution of the grounding line position over time. (c) Evolution of the grounding line position overlaid on a three-dimensional view of the bedrock elevation. Solid colour lines in (b) and (c) indicates the position of the DInSAR grounding lines between 1992 and present, and are colour coded on a scale from white (1992) to red (2021).





**Figure 2.** Selection of double difference SAR interferograms of Petermann Glacier, Greenland, collected between 1992 and
2021. Solid light green lines indicate the digitized grounding line positions for a given year.



**Figure 3.** Relative grounding line position and change in ice speed of Petermann glacier. a) Relative grounding line position
with respect to the location in February 1992 for four locations across Peterman's width (see profiles T1 to T4 in Fig. 1b).





Slopes of each linear fit (indicating the rate of grounding line retreat) are shown in the same colour as each line. b) Long-term

change in ice flow velocity at the grounding line, c) and d) Hovmöller diagram of velocity at the cross sections A-A' and B-

B' (see Figure 1a). e) Percentage change in speed relative to the 2014 winter average along profile C-C'. The vertical lines

indicates the position of the 1992 and 2021 grounding lines on the profile C-C'.



**Figure 4.** Shear strain rates of Petermann Glacier in 2000 (a) and 2019 (b), overlaid on Landsat images from the same year.

Strain rates are colour coded from violet to green. (c) and (d) display the surface expression of the shear strain maps from

420  Landsat-7 and Landsat-8 images taken in 2000 and 2019, respectively. Solid colour lines shows the grounding line position

between 1992 and present, colour coded on a scale from white (1992) to red (2021).





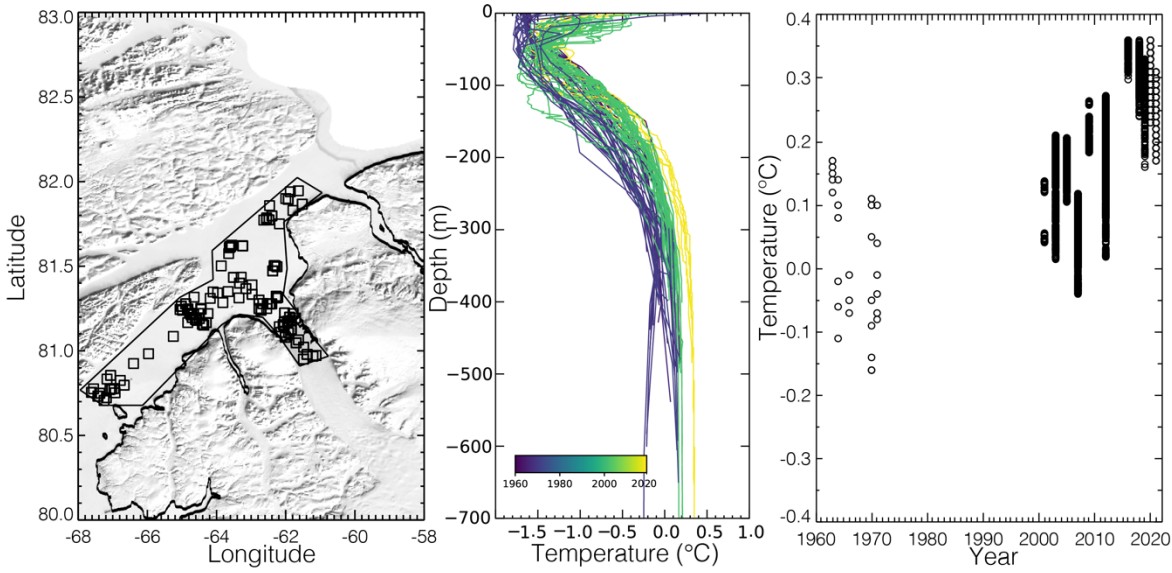

**Figure 5.** Conductivity, Temperature, Depth (CTD) measurements near Petermann Glacier, Greenland with a) CTD location from 1963-2021 colour coded by years as in b) with coastline in black and region of extraction in thin black; b) potential temperature from CTD casts coded from 1963-2021; and c) change in temperature averaged between 350 and 450 m depth from 1963 to 2021 colour coded by years.