# Peer review of "Ongoing grounding line retreat and fracturation initiated at the Petermann Glacier ice shelf, Greenland after 2016"

_The Cryosphere, 2022_

## Referee Comment (RC2)

**Review of Millan et al. Ongoing grounding line retreat and fracturation initiated at the Petermann Glacier ice shelf, Greenland after 2016**

March 2022

**General comments**

The authors present a new extended record of grounding line migration and ice velocity for Petermann Glacier and provide insight into the causes of recent retreat and the impact of ice shelf fracturing on the flow of the glacier. They conclude that recent retreat is likely to be as a result of an increase in ocean temperatures in the fjord, but that retreat of the grounding line is modulated by the bed topography. In general the paper is well structured and written and the figures are appropriate. I have some more general comments outlined below prior to publication, followed by some specific line comments.

The introduction needs to include a clearer statement of the justification for the research. Presumably including something about how newly available data make it possible to provide an updated record of grounding line movement to better understand dynamics of Petermann Glacier. You could then better link this to previous observations of calving events, and state how while recent calving events have been observed, there is no recent data on grounding line movement and ice flow speeds over the last 10 years. In addition, I was looking for a better explanation and justification for expanding on the previous work by Hogg et al., 2016. It would be good in general to make a better comparison between your record and theirs in the results or discussion.

I was left a little confused about the timing of events you are presenting. In the first line of the discussion you suggest the speed-up preceded the grounding line retreat in 2017. However, the abstract to me suggests the opposite, saying 'as a result, the glacier sped up'. Again in the discussion you state that the velocity remained stable after 2018 despite continued grounding line retreat. In general, the sequence of events needs to be made clearer, and be consistent throughout the manuscript.

There are a number of a different datasets from different time periods that have been integrated into the paper. I think it would be helpful for the reader if you were to include a table or diagram that gives a complete overview of all the datasets used, the time spans and how many measurements were acquired from each.

I found the use of the phrase 'stable' throughout the paper to be unclear, especially given that at present surface thinning inland of the grounding line is on the order of a metre per year (as stated on line 202 of the Discussion). Uses of the term stable need to be qualified throughout, or at least include a clear definition at the first instance.

I have a number of specific line comments (detailed below) that will hopefully help to improve the clarity of the paper.

**Specific comments**

**Line 19:** Here you state 'since 2017' but the title says 'after 2016'. It would be better if these were consistent. E.g. 'initiated in 2017' or 'After 2016 the grounding line began to retreat...'

**Line 20:** I guess this 5km is part of the 7 km stated in the previous sentence, but this could be clearer. For example saying 'the majority of retreat in the central sector took place between 2017 and 2021 (5 km) '

**Line 21:** State the time frame over which these fractures developed. Also, state the direction (along/across flow) in which the ice shelf is split into three sections.

**Line 22:** Do you mean as a result of the ocean warming, or as a result of the grounding line retreat?

**Line 30:** It would be useful to include a sentence to explain the process by which ice shelf weakening would destabilise the sector and lead to an increase in sea level rise.

**Line 34** How is this record over the last 20-years if the paper was published in 2019. Surely it was the 20 years prior to 2019? State the full time period over which this increase occurred. Also are there some error estimates associated with these ice discharge estimates? Is a 1 Gt/yr difference beyond those?

**Line 40:** I think the phrase 'were not detectable' could be explained more clearly. State the reason why, and then at some stage how you overcome the challenges/expand on their previous work, given that you cover a similar time period to these studies.

**Line 46:** This paragraph feels disjointed from the rest of the introduction. See general comment on providing a clear justification on extending this record. You do hint at it, but I think a clear statement at the start of this paragraph should be included to make a clear link between the justification of extending the record, the available data, and why it is of interest to measure grounding line migration.

**Line 64:** This subheading might be better as grounding line positions.

**Section 2.1:** I found it difficult to get a clear sense of the number of scenes used to obtain grounding line positions. I suggest including a table of all the data types and dates to provide an overview. I also think that a statement in this section comparing the data/method you use in comparison to Hogg et al., 2016 is necessary here, and importantly how you have expanded upon their existing record.

**Line 83:** This paragraph needs a better starting sentence. Consider moving the final sentence of the paragraph to the start with something along the lines of 'To complement the Sentinel-1 record we use...'

**Line 91:** Restate the epochs.

**Line 109:** Why did you choose these two time stamps (2000 and 2019) to calculate the strain rates?

**Line 127-128:** I'm still unsure exactly how many scenes/grounding line measurements you have between 1992 and 2011? Is it just 3 ('92,'96','11) and in which case is that enough to say that it was stable during this period. Also 'relatively stable' is ambiguous, something like 'has not advanced or retreated substantially from its position in 1992' might be better.

**Line 128:** Make a more quantitative statement about the errors. How did you calculate the uncertainties in your grounding line positions, how did you arrive at 1 km? An error estimate and explanation is needed either here or in the methods (Section 2.1).

**Line 129:** Can you be confident about a 1.1 km retreat being a true signal if you state in the previous sentence that the uncertainties are 'about 1 km'? Error estimates for your grounding line measurements are needed.

**Line 152:** Why not also describe here the regions of apparent fracture on the eastern section of the tongue near to the calving front. Could these have also decoupled these sections of the lower ice

shelf from the flow? At the very least you could make a comparison to the findings of Ruckamp et al.. 2019 in the discussion when you mention the 'partially-decoupled flow regime' (Line 190).

**Line 155:** Nowhere in Section 2.2. did you mention the velocity record was extended back to 1984. Also data from 1984 is not shown on Fig. 3b (x-axis starts at 1992), so why mention it here at all?

**Line 155:** Fig. 3c shows a profile approx 20 km long. Do you mean C-C′ here?

**Line 157:** This statement 'glacier motion was fairly stable' has to be quantified or reworded, perhaps along the lines of, 'there was very little variability in ice speeds (+/- x m yr-1) between xxx and 2014'.

**Line 163:** Where is the evidence for this statement? Have you compared summer speed-ups to the melt season?

**Line 176-177:** This statement is ambiguous, what constitutes a warming signal and what is a strong signal? Clarify.

**Line 177:** Last decade from when? Today? or the last decade of the 1970-2000s record? Include the time period for 'the last decade'.

**Line 179:** Only during the 2000s? or also during the 2010s? State the full time period that you are referring to.

**Line 190:** Can you provide some extra detail on what this term 'partially-decoupled flow regime' means and what is the criteria and what the impact may be on the flow upstream of the grounding line?

**Line 192:** What constitutes a 'destabilised glacier'? Is it one that formerly had an ice shelf/tongue, that has since disappeared. A definition is needed.

**Line 194:** Be specific here. What is it about your results exactly that suggests the glacier might be destabilising or weakening. Can you also make some statements on the implications of this? If it is has already decoupled from the flow inland of the grounding line, would future 'destabilisation' of the ice shelf have an impact on ice speeds and grounding line retreat? I think the Discussion in general needs to cover some previous studies (that have been mentioned in the introduction) on this decoupling/impact of calving events/thinning on the ice shelf and put into the context of the results presented here.

**Line 212:** I don't think 'coincides' makes sense here, because the bed topography is not a temporal dataset. Consider rephrasing along the lines of '...migration occurred across a section of retrograde bed topography...'

**Line 213:** Perhaps some additional examples, of outlet glaciers in Greenland (albeit without floating ice tongues) would be useful here, e.g. Humboldt Glacier (Carr et al., 2015), or Tracy/Heilprin Glaciers (Porter et al., 2014).

**Line 218:** Perhaps qualify that this retreat would not purely be determined by the bedrock alone, but buttressing plays a role in the stability of the grounding line on a retrograde bed slope (e.g. Gudmundsson et al., 2012).

**Line 222:** 'first time and quite recently' is ambiguous. Rephrase to something along the lines of 'In the last 5 years (2017-2021) the grounding line has begun to retreat more rapidly than in the previous 25'

**Line 225:** State the timeframe.

**Figure 1:** What is the purpose of profile D-D'? It does not appear to be shown in Figure 3 nor referred to in the text.

**Figure 2:** Add labels to the panels in this figure, as you refer to Fig. 2a etc in Section 3.2. The caption for this figure is incomplete. Also, you mention the 'decorrelation sturctures' in the text and refer to Fig.2 so it would be useful to add arrows and labels to point to the reader to the exact

locations of these structures.

**Figure 3:** a) If the grounding line positions are with respect to their position in Feb 1992, why do some of the points shown for Feb 1992 fall below zero? Surely if Feb 1992 is the reference, all points at this start point should equal zero. b) it is quite difficult to see the trend in the velocity data. Perhaps it would be useful to put on the annual or winter average ontop? c) The x-axis scale starts off in intervals of 5 up to 10 and then jumps to 19. Is this a mistake?

**Figure 4:** The 1992 line in the legend cannot be seen on the white background (this is the same for Fig.1). Either change the legend background or the colour of the line. Also the white lines are difficult to see on the white background of the strain rates in a) and b). Change 'present' to 2021.

**Figure 5:** You refer to a),b) and c) in the caption but they are not shown on the panels. Also the caption is incorrect, neither a) or c) show data colour coded by years. Add location labels to the first panel, e.g. Petermann Glacier, Petermann Fjord and Nares Strait, all of which are places you refer to in the text but the reader is not shown where they are.

**Technical corrections**

**Line 59:** Sometimes you capitalise Glacier (after Petermann) and other times not. Needs to be the same throughout.

**Line 74:** Perhaps start a new paragraph at 'For the time period 2014 to 2021'

**Line 104:** Amend ERS-1/2

**Line 105:** There is no year associated with the Joughin and Howat reference

**Line 157:** I think this should be Fig. 3b

---

## Author Response (AR1)

Dear Dr Stef Lhermitte,

On behalf of my co-authors, I am pleased to resubmit our manuscript entitled "Ongoing grounding line retreat and fracturation initiated at the Petermann Glacier ice shelf, Greenland after 2016" after minor revisions, for publication in The Cryosphere.

We answered all comments and more specifically we clarified the Discussion section and the timing of events that are happening at Petermann Glacier. Moreover we also provide an error assessment for the grounding line retreat. Please note that we have removed the sentence stating that fractures were across the entire thickness of Petermann ice shelf, since we could not find proper radar measurements to bring evidence of this.

You will find attached the manuscript, five figures, along with the tracked changes and response to all specific comments from Reviewer #1 and #2. We sincerely hope that you will find this paper suitable for publication in The Cryosphere.

Please do not hesitate to reach out should you have any questions about our submission. Best regards,

On behalf of all authors,

Dr. Romain Millan

Response to Reviewer #1

The authors present an interesting combination of data to visualize the timeline of the Petermann glacier's retreating groundling line, which fits within the scope of TC. The novelty of the research is the long time series of grounding line mapping which has not yet been done on the Petermann glacier. However, the methods used in the study are not very novel and much of the reported findings are a temporal extension of other studies. Based on the findings the conclusion is that the grounding line is retreating following an increase in the ocean water temperature in the nearby Naires Straight. The authors could improve on the clarity of the conclusion by providing a clear timeline of events with an explanation of the causality that precedes them. Additionally, the conclusions would carry more weight when an error estimation is provided with each result. Therefore, with some minor revision the paper could be published.

 General comment:

As stated earlier a more clear timeline (perhaps even with an actual timeline) would immediately get the message of the paper across. Additionally, considering the signal-to-noise was too low in earlier studies to visualize grounding line migration, a confidence interval has to be provided with the reported findings.

We would like to thank Reviewer #1 for his insightful comments on our manuscript.

We now provide error bars for the relative position of the grounding lines (see later response to comment), and we have extensively revised the discussion to provide a clearer timeline of the recent events affecting Petermann.

Specific comments:

Abstract:

L22: As a result of the warming or the grounding line retreat? Rückamp et al. (2019) suggest that the speed-up is due to calving. This in turn can be due to an increase of ocean water temperature, but is a bit more complicated (see Shroyer et al. *"Seasonal control of Petermann Gletscher ice-shelf melt by the ocean's response to sea-ice cover in Nares Strait,"* 2017).

Authors: As a result of the grounding line retreat. We clarified the sentence accordingly.

L24: change "accentuate" to speed-up/enhance

Authors: Done

From L32-45 From the text it seems there is no reason at all to do this study, as everything appears to be stable. Therefore, the reason for this study should be stated more clearly: i.e. "earlier it was not possible to map the grounding line migration, due to a low signal-to-noise ratio. Considering the importance of grounding line migration to glacier stability we now show its evolution and provide additional data to explain its behaviour".

Authors: Grounding line mapping was possible in the past, and we can show it with the dense coverage of ERS-1/2 double difference interferograms in 1992 (Figure 2). These "historical" data are optimal for grounding line mapping, with 1-3 days repeat cycles, hence with a good signal to noise ratio. However, in the past, the grounding line migration of Petermann could not be distinguished from the signal due to tides (Hogg et al., 2016), as it did not exceed 500-600 m, hence the glacier was considered dynamically stable. What is new here, is that the grounding line position of Petermann significantly changed, exceeding its natural variations due to the tidal signal. In short, Petermann has clearly entered a phase of retreat similar to that observed for tidewater glaciers elsewhere in Greenland or Antarctica. We believe this is an important event that should be reported to the scientific community: a major Greenland glacier that was dynamically stable between 1992 and 2011 has entered a phase of retreat and acceleration since about 2015. We now specify this more clearly at L56.

L46-57: This is data/method description and therefore doesn't fit in the introduction. Or if this is new data that now allows you to do this study then mention it shortly.

Authors : This is already stated clearly in the first sentence of the paragraph: "Since 2011, the launch of a constellation of two Sentinel-1 […] satellites have increased the perspective to study glacier dynamics and grounding line migration" and later on at line 51 "This continuous

observation record has made it possible to map the grounding line again since ERS-1, but on multiple occasions, several times a year, hence providing information about the zone of short-term migration or grounding zone.". No change.

L59-60: Do you combine these data sources or do you use them separately for two different things?

Authors: We think that the reviewer is referring to the sentence about Cosmo-SkyMed dataset at L55 of the original manuscript. We modified the sentence to clarify that we combine both Sentinel-1 and Cosmo-SkyMed data to map the grounding line at L66-70.

L61: expand a bit on how you get to those conclusions. Just showing data is not enough, you also need to explain how they are related and what explanatory features they provide.

Authors: We revised the discussion extensively to better explain how data are related to each other, and what processes may explain those connections.

L109-117: Note that it is not strain, but stress that causes ice shelfs to fracture. Strain is of course related to stress, but ice rheology is an important factor as well. Additionally note that movement of decoupled ice does not suggest high strain rates, but just shows relative movement.

Authors: Agreed. We clarified the sentence accordingly at L128.

L135: Can you include some form of confidence interval for the estimations of the grounding line retreat?

Authors: Mohajerani et al., 2021 quantified, at the scale of Antarctica, uncertainties on grounding line delineation, and calculated a MAD of 110 m and an IQR of 155 m. Error bars were added accordingly to Figure 3a. We also provide $R^2$ values and standard errors on the slopes for the linear fits that are used in Figure 3a.

L163-164: Do you have data or reference supporting this statement?

Authors: Agreed. The sentence was removed.

L187: You state that an increase in flow leads to enhanced thinning which causes the grounding line to retreat. Although there is some truth in that statement it would be better to rephrase it a bit, as grounding line retreat is also related to the fjord topography. When warm water can flow along a retrograde slope the grounding line will retreat much faster than when presented with a prograde slope. Additionally I can imagine a scenario where a reduction in buttressing increases the discharge of the glacier and causes an advancement of the grounding line. One not necessarily follows the other.

Authors: Agreed. The paragraph was rephrased accordingly at L215-254 of the tracked change file, and we move the last paragraph of the discussion, which was discussing specifically the retrograde bed slope at L254-264, to have a more coherent discussion.

L197: Can you explain a bit more about this model, so the reader does not have to read the paper of Rignot et al. (2016)?

Authors: Agreed. A sentence was added at L275 to describe the simple model used by Rignot et al. (2016).

L199: Is this $q_m$ part of a formula? If so, please provide the whole formula.

Authors: We removed the mention of $q_m$ as it was not necessary for the discussion.

L200-205: Considering you are an order of magnitude off the observed grounding line retreat, is this a really useful calculation to make? Indeed subglacial water discharge can influence the basal melt rate, but make sure to provide how much this effect can be. Same for pressurized seawater. If these effects can explain the discrepancy between observed and calculated than your argument for making the calculation is a lot stronger.

Authors: We revised extensively the discussion and specifically this paragraph to provide more clarity on this calculation and the related processes. See also response to reviewer #2.

L218: consider replacing "accentuate" with promote/increase/further

Authors: Agreed. We replaced "accentuate" with "promote".

Conclusion:

The conclusion is basically repeating the abstract. Consider leaving it out as it does not add anything to the paper and allows for more space to explain the methods a bit further.

Authors: Manuscript guidelines from the cryosphere requires to have a "Conclusions" section. Furthermore, we have revised the conclusions so that it does not repeat itself in comparison with the abstract.

FIG1:

Where is cross section D-D' used? Pane 1c is quite difficult to interpret and doesn't make the situation more clear than 1b. Consider replacing it by an along fjord cross section. Here as well some notion of confidence interval with the visualized data would help.

Authors: We removed the Panel 1c accordingly. The confidence interval was added to the relative grounding line position plot, in Figure 3a, which is more appropriate.

Response to Reviewer #2

General comments

The authors present a new extended record of grounding line migration and ice velocity for Petermann Glacier and provide insight into the causes of recent retreat and the impact of ice shelf fracturing on the flow of the glacier. They conclude that recent retreat is likely to be as a result of an increase in ocean temperatures in the fjord, but that retreat of the grounding line is modulated by the bed topography. In general the paper is well structured and written and the figures are appropriate. I have some more general comments outlined below prior to publication, followed by some specific line comments.

The introduction needs to include a clearer statement of the justification for the research. Presumably including something about how newly available data make it possible to provide an updated record of grounding line movement to better understand dynamics of Petermann Glacier. You could then better link this to previous observations of calving events, and state how while recent calving events have been observed, there is no recent data on grounding line movement and ice flow speeds over the last 10 years. In addition, I was looking for a better explanation and justification for expanding on the previous work by Hogg et al., 2016. It would be good in general to make a better comparison between your record and theirs in the results or discussion.

I was left a little confused about the timing of events you are presenting. In the first line of the discussion you suggest the speed-up preceded the grounding line retreat in 2017. However, the abstract to me suggests the opposite, saying 'as a result, the glacier sped up'. Again in the discussion you state that the velocity remained stable after 2018 despite continued grounding line retreat. In general, the sequence of events needs to be made clearer, and be consistent throughout the manuscript.

There are a number of a different datasets from different time periods that have been integrated into the paper. I think it would be helpful for the reader if you were to include a table or diagram that gives a complete overview of all the datasets used, the time spans and how many measurements were acquired from each.

I found the use of the phrase 'stable' throughout the paper to be unclear, especially given that at present surface thinning inland of the grounding line is on the order of a metre per year (as stated on line 202 of the Discussion). Uses of the term stable need to be qualified throughout, or at least include a clear definition at the first instance.

I have a number of specific line comments (detailed below) that will hopefully help to improve the clarity of the paper.

Authors: We would like to thank Reviewer #2 for the insightful comments. In response to these, we have revised the Discussion section, where we work on a clearer understanding of the events happening at Petermann Glacier. As it was suggested by Reviewer #2, this new discussion is now built with a thorough comparison to the findings of the 2019 Rückamp et al. study. We now discuss the changes in dynamic and its relation to loss in buttressing due to calving and fracturation.

Specific comments

Line 19: Here you state 'since 2017' but the title says 'after 2016'. It would be better if these were consistent. E.g. 'initiated in 2017' or 'After 2016 the grounding line began to retreat...'

Authors: Agreed.

Line 20: I guess this 5km is part of the 7 km stated in the previous sentence, but this could be clearer. For example saying 'the majority of retreat in the central sector took place between 2017 and 2021 (5 km) '

Authors: Agreed. The sentence was modified accordingly L19.

Line 21: State the time frame over which these fractures developed. Also, state the direction (along/across flow) in which the ice shelf is split into three sections.

Authors: Agreed. The sentence was modified accordingly L21.

Line 22: Do you mean as a result of the ocean warming, or as a result of the grounding line retreat?

Authors: The sentence was modified accordingly to make the timing of events and chain of causality clearer, in response to reviewer #1.

Line 30: It would be useful to include a sentence to explain the process by which ice shelf weakening would destabilize the sector and lead to an increase in sea level rise.

Authors: Agreed. A sentence was added at L36-39.

Line 34 How is this record over the last 20-years if the paper was published in 2019. Surely it was the 20 years prior to 2019? State the full time period over which this increase occurred. Also are there some error estimates associated with these ice discharge estimates ? Is a 1 Gt/yr difference beyond those ?

Authors: Agreed. We added the full time period at line 42-43 along with the error on the ice discharge. This increase in ice discharge is indeed close from the error bars. Hence we mitigate our sentence by rephrasing to "Since 1995, this glacier seems to have increased its ice discharge from $10.6 \pm 1$ Gt/yr up to $11.7 \pm 1.2$ Gt/yr in 2018 (Mouginot et al., 2019)."

Line 40: I think the phrase 'were not detectable' could be explained more clearly. State the reason why, and then at some stage how you overcome the challenges/expand on their previous work, given that you cover a similar time period to these studies.

Authors: Agreed. We clarified the sentence at L47-51.

Line 46: This paragraph feels disjointed from the rest of the introduction. See general comment on providing a clear justification on extending this record. You do hint at it, but I think a clear

statement at the start of this paragraph should be included to make a clear link between the justification on extending the record, the available data, and why it is of interest to measure grounding line migration.

Authors: Agreed. We added a sentence at L56.

Line 64: This subheading might be better as grounding line positions.

Authors: Agreed. Subheading was modified accordingly.

Section 2.1: I found it difficult to get a clear sense of the number of scenes used to obtain grounding line positions. I suggest including a table of all the data types and dates to provide an overview. I also think that a statement in this section comparing the data/method you use in comparison to Hogg et al., 2016 is necessary here, and importantly how you have expanded upon their existing record.

Authors: We provide a supplementary table with all synthetic aperture radar data that were manually digitized in this study. We added a reference L109, 143.

Line 83: This paragraph needs a better starting sentence. Consider moving the final sentence of the paragraph to the start with something along the lines of 'To complement the Sentinel-1 record we use...'

Authors. Agreed. The sentence was moved to the beginning of the paragraph accordingly.

Line 91: Restate the epochs.

Authors. The dates are provided as part of the Table (see previous comment). The sentence was modified accordingly L109.

Line 109: Why did you choose these two time stamps (2000 and 2019) to calculate the strain rates?

Authors: We choose 2000 and 2019 to cover time periods before and after the grounding line retreat. Year 2019 was the latest available year in ice velocity processing, and year 2000 the earlier year with the best quality. The point of this figure is to illustrate the strong changes in strain rate, while the full timeline of fracture propagation is shown in Figure 2. A sentence was added at line 129-131.

Line 127-128: I'm still unsure exactly how many scenes/grounding line measurements you have between 1992 and 2011? Is it just 3 ('92,'96','11) and in which case is that enough to say that it was stable during this period. Also 'relatively stable' is ambiguous, something like 'has not advanced or retreated substantially from its position in 1992' might be better.

Authors: A table on the amount of grounding line measurements are provided as a response to an earlier comment. Considering "relatively stable", we have changed the sentence accordingly at line 151.

Line 128: Make a more quantitative statement about the errors. How did you calculate the uncertainties in your grounding line positions, how did you arrive at 1 km? An error estimate and explanation is needed either here or in the methods (Section 2.1).

Authors: Error bar on the relative grounding line position was added to Figure 3a along with standard error on the slopes and $R^2$ values on the linear fits, as a response to Reviewer #1. In addition, we modified the sentence at L150-153, to provide a quantification on the uncertainty. We also modified the paragraph at L154 to clarify our point here about the stability of Petermann in 1992-2011.

Line 129: Can you be confident about a 1.1 km retreat being a true signal if you state in the previous sentence that the uncertainties are 'about 1 km'? Error estimates for your grounding line measurements are needed.

Authors: This point is covered with the previous comment.

Line 152: Why not also describe here the regions of apparent fracture on the eastern section of the tongue near to the calving front. Could these have also decoupled these sections of the lower ice shelf from the flow? At the very least you could make a comparison to the findings of Ruckamp et al.. 2019 in the discussion when you mention the 'partially-decoupled flow regime' (Line 190).

Authors: We revised the discussion section L215-230 and provide further discussion about the results and comparison with Rückamp et al., 2019.

Line 155: Nowhere in Section 2.2. did you mention the velocity record was extended back to 1984. Also data from 1984 is not shown on Fig. 3b (x-axis starts at 1992), so why mention it here at all?

Authors: Agreed. We have a velocity record starting in 1984 (c.f. Mouginot et al. 2019) but show only back to 1992. Accordingly, we changed 1984 by 1992 in the text L182.

Line 155: Fig. 3c shows a profile approx 20 km long. Do you mean C-C′ here?

Authors: Yes we were referring to C-C'. The reference to the figure was corrected accordingly.

Line 157: This statement 'glacier motion was fairly stable' has to be quantified or reworded, perhaps along the lines of, 'there was very little variability in ice speeds (+/- x m yr-1) between xxx and 2014'.

Authors: Agreed. The sentence was modified at L184.

Line 163: Where is the evidence for this statement? Have you compared summer speed-ups to the melt season?

Authors: Agreed. We removed this sentence accordingly as we do not compare our time series with the history of the melt season.

Line 176-177: This statement is ambiguous, what constitutes a warming signal and what is a strong signal? Clarify.

Authors: Agreed. The sentence was modified at L205-215 to better define the ranges at which ocean temperature have warmed.

Line 177: Last decade from when? Today? or the last decade of the 1970-2000s record? Include the time period for 'the last decade'.

Authors: Agreed. Precise time period was added at L210.

Line 179: Only during the 2000s? or also during the 2010s? State the full time period that you are referring to.

Authors: Agreed. We added precise dates for this sentence L212.

Line 190: Can you provide some extra detail on what this term 'partially-decoupled flow regime' means and what is the criteria and what the impact may be on the flow upstream of the grounding line?

Authors: Agreed. Some details on the description of the decoupled flow regime along with the development of fractures were added at L198-205.

Line 192: What constitutes a 'destabilised glacier'? Is it one that formerly had an ice shelf/tongue, that has since disappeared. A definition is needed.

Authors: Agreed. Here we mean a glacier that experiences increased rates of mass losses. We clarified our sentence at line 285.

Line 194: Be specific here. What is it about your results exactly that suggests the glacier might be destabilising or weakening. Can you also make some statements on the implications of this?

Authors: The Discussion section was restructured according to a General comment of Reviewer #1, in order to make our conclusions clearer. Details about the weakening and destabilization of the glacier are present at L215-250.

If it is has already decoupled from the flow inland of the grounding line, would future 'destabilisation' of the ice shelf have an impact on ice speeds and grounding line retreat? I think the Discussion in general needs to cover some previous studies (that have been mentioned in the introduction) on this decoupling/impact of calving events/thinning on the ice shelf and put into the context of the results presented here.

Authors: Agreed. More details on the succession of event leading to ice shelf fracturation, grounding line retreat and ice velocity are given at L215-250.

Line 212: I don't think 'coincides' makes sense here, because the bed topography is not a temporal dataset. Consider rephrasing along the lines of '...migration occurred across a section of retrograde bed topography...'

Authors: Agreed. This was modified accordingly at line 256.

Line 213: Perhaps some additional examples, of outlet glaciers in Greenland (albeit without floating ice tongues) would be useful here, e.g. Humboldt Glacier (Carr et al., 2015), or Tracy/Heilprin Glaciers (Porter et al., 2014).

Authors: Agreed. The sentence was modified accordingly at L258.

Line 218: Perhaps qualify that this retreat would not purely be determined by the bedrock alone, but buttressing plays a role in the stability of the grounding line on a retrograde bed slope (e.g. Gudmundsson et al., 2012).

Authors: The role of buttressing is now discussed at L215-240.

Line 222: 'first time and quite recently' is ambiguous. Rephrase to something along the lines of 'In the last 5 years (2017-2021) the grounding line has begun to retreat more rapidly than in the previous 25'

Authors: Agreed. We have restructured the conclusion in response to a suggestion from Reviewer #1.

Line 225: State the timeframe.

Authors: Agreed. We have restructured the conclusion in response to a suggestion from Reviewer #1.

Figure 1: What is the purpose of profile D-D'? It does not appear to be shown in Figure 3 nor referred to in the text.

Authors: Profile D-D' was removed from Figure 1.

Figure 2: Add labels to the panels in this figure, as you refer to Fig. 2a etc in Section 3.2. The caption for this figure is incomplete. Also, you mention the 'decorrelation structures' in the text and refer to Fig.2 so it would be useful to add arrows and labels to point to the reader to the exact locations of these structures.

Authors: Agreed. The figure was modified accordingly.

Figure 3: a) If the grounding line positions are with respect to their position in Feb 1992, why do some of the points shown for Feb 1992 fall below zero? Surely if Feb 1992 is the reference, all points at this start point should equal zero. b) it is quite difficult to see the trend in the velocity data. Perhaps it would be useful to put on the annual or winter average ontop? c) The x-axis scale starts off in intervals of 5 up to 10 and then jumps to 19. Is this a mistake?

Authors: We have 8 grounding line measurements in 1992, spanning from January to March, which explains why all points are not equal to zero. Figure 3b is used to discuss the amplitude of the seasonal variation in surface flow velocity. However, the long-term trend and relative changes are shown in Panel c, d and e, hence we do not think it would add more information to add annual average on top.

Figure 4: The 1992 line in the legend cannot be seen on the white background (this is the same for Fig.1). Either change the legend background or the colour of the line. Also the white lines are difficult to see on the white background of the strain rates in a) and b). Change 'present' to 2021.

Authors: Agreed. We changed the background of the legend, and the color scale of the strain rates in order to improve readability. We also changed present to 2021.

Figure 5: You refer to a),b) and c) in the caption but they are not shown on the panels. Also the caption is incorrect, neither a) or c) show data colour coded by years. Add location labels to the first panel, e.g. Petermann Glacier, Petermann Fjord and Nares Strait, all of which are places you refer to in the text but the reader is not shown where they are.

Authors: Agreed. We added location label and adjusted the legend accordingly.

Technical corrections
Line 59: Sometimes you capitalise Glacier (after Petermann) and other times not. Needs to be the same throughout.

Authors: We capitalized Glacier throughout the entire manuscript for consistency.

Line 74: Perhaps start a new paragraph at 'For the time period 2014 to 2021'

Authors: Done.

Line 104: Amend ERS-1/2

Authors: Done.

Line 105: There is no year associated with the Joughin and Howat reference

Authors: Corrected.

Line 157: I think this should be Fig. 3b

Authors: Agreed.

---

## Author Response (AR2)

Dear Dr Stef Lhermitte,

On behalf of my co-authors, I am pleased to resubmit our manuscript entitled "Ongoing grounding line retreat and fracturation initiated at the Petermann Glacier ice shelf, Greenland after 2016" after minor revisions, for publication in The Cryosphere.

We answered all the remaining comments from the reviewers, including on the timing of events in the Discussion section and reclarified some parts of the text.

You will find attached the manuscript, five figures, along with the tracked changes and response to all specific comments from Reviewer #1 and #2.

Please do not hesitate to reach out should you have any questions about our submission. Best regards,

On behalf of all authors,

Dr. Romain Millan

Response to Reviewer #1

The authors improved the manuscript and is ready for publication after two minor technical corrections:

Authors: We would like to thank Reviewer #1 in helping to improve the quality of our manuscript throughout two rounds of revisions.

L48: the word "lost" appears to be missing.

Authors: Done.

References: "Flament & Rémy 2017", cited in L224, is missing in the References list.
Authors: We added the reference accordingly at L300

Response to Reviewer #2

I thank the authors for addressing my previous comments and for their hard work improving the manuscript. I particularly appreciate the clarification on the timing of the events and the improvements made to the Discussion. I have just a few additional comments, but otherwise I am happy to recommend the manuscript be published.

Authors: We would like to thank Reviewer #2 in helping to improve the quality of our manuscript. We have addressed the remainder of the concerns below.

Minor comments:

Line 22: Change to "followed" or "occurred after"

Authors: Done.

Line 53: "Has" instead of "have"

Authors: Done.

Line 81: Put the reference for the GIMP v1 DEM as it is the first time it is introduced. Also, Howat et al. 2017 doesn't appear in the reference list.

Authors: Done. We have added the reference to GIMP v1 (Howat et al 2014 at L82), and added Howat et al., 2017 to the reference list L305-310.

Line 92: Perhaps change "complete" to "complement"
Authors: Done.

Line 198: Can you improve the link between these sentences? It wasn't fully clear to me how constant velocities between 1986 and 2010 are then consistent with the results of Rückamp et al. 2019 presented after 2012.
Authors: We moved this sentence to the second paragraph of the discussion, where it belonged initially (see L211-212).

Line 207: Can you clarify again what may have caused the "loss of ice shelf resistance" and re-state the timeframe. I'm also not sure the Gudmundsson reference is correct here (see following comment).
Authors: We have clarified that the cause of the loss of buttressing was due to the 2012 calving event (L209), and restated the dates of the events at L209-210. The reference to Gudmundson was removed from the sentence (see next comment).

Line 222: The caveat (ice shelf buttressing) to bed slope being the only control on future retreat (Gudmundsson et al. 2012) I think would be better be added to this third paragraph with some additional explanation.
Authors: We have added the reference to Gudmundson at L226-228, along with an explanation on why the current conditions at Petermann may promote future retreat.

Line 238: Perhaps reiterate why rising ocean temperatures can lead to retreat of the grounding line here.
Authors: We clarified this sentence at L237.

Line 240: Is there a reference for this statement that ocean temperatures may have caused large calving events in 2010 and 2012?

Authors: The sentence was removed accordingly.